# Learning Treewidth-Bounded Bayesian Networks with Thousands of Variables

**Mauro Scanagatta**
IDSIA*, SUPSI†, USI‡
Lugano, Switzerland
mauro@idsia.ch

**Giorgio Corani**
IDSIA*, SUPSI†, USI‡
Lugano, Switzerland
giorgio@idsia.ch

**Cassio P. de Campos**
Queen's University Belfast
Northern Ireland, UK
c.decampos@qub.ac.uk

**Marco Zaffalon**
IDSIA*
Lugano, Switzerland
zaffalon@idsia.ch

## Abstract

We present a method for learning treewidth-bounded Bayesian networks from data sets containing thousands of variables. Bounding the treewidth of a Bayesian network greatly reduces the complexity of inferences. Yet, being a global property of the graph, it considerably increases the difficulty of the learning process. Our novel algorithm accomplishes this task, scaling both to large domains and to large treewidths. Our novel approach consistently outperforms the state of the art on experiments with up to thousands of variables.

## 1 Introduction

We consider the problem of structural learning of Bayesian networks with bounded treewidth, adopting a score-based approach. Learning the structure of a bounded treewidth Bayesian network is an NP-hard problem (Korhonen and Parviainen, 2013). Yet learning Bayesian networks with bounded treewidth is necessary to allow exact tractable inference, since the worst-case inference complexity is exponential in the treewidth $k$ (under the exponential time hypothesis) (Kwisthout et al., 2010).

A pioneering approach, polynomial in both the number of variables and the treewidth bound, has been proposed in Elidan and Gould (2009). It incrementally builds the network; at each arc addition it provides an upper-bound on the treewidth of the learned structure. The limit of this approach is that, as the number of variables increases, the gap between the bound and the actual treewidth becomes large, leading to sparse networks. An *exact* method has been proposed in Korhonen and Parviainen (2013), which finds the highest-scoring network with the desired treewidth. However, its complexity increases exponentially with the number of variables $n$. Thus it has been applied in experiments with 15 variables at most. Parviainen et al. (2014) adopted an anytime integer linear programming (ILP) approach, called TWILP. If the algorithm is given enough time, it finds the highest-scoring network with bounded treewidth. Otherwise it returns a sub-optimal DAG with bounded treewidth. The ILP problem has an exponential number of constraints in the number of variables; this limits its scalability, even if the constraints can be generated online. Berg et al. (2014) casted the problem of structural learning with limited treewidth as a problem of weighted partial Maximum Satisfiability. They solved the problem exactly through a MaxSAT solver and performed experiments with 30 variables at most. Nie et al. (2014) proposed an efficient anytime ILP approach with a polynomial number of constraints

in the number of variables. Yet they report that the quality of the solutions quickly degrades as the number of variables exceeds a few dozens and that no satisfactory solutions are found with data sets containing more than 50 variables. Approximate approaches are therefore needed to scale to larger domains.

Nie et al. (2015) proposed the method S2. It exploits the notion of k-tree, which is an undirected maximal graph with treewidth $k$. A Bayesian network whose moral graph is a subgraph of a k-tree has treewidth bounded by $k$. S2 is an iterative algorithm. Each iteration consists of two steps: a) sampling uniformly a k-tree from the space of k-trees and b) recovering a DAG whose moral graph is a sub-graph of the most promising sampled k-tree. The goodness of the k-tree is assessed via a so-called *informative score*. Nie et al. (2016) further refine this idea, obtaining via A* the k-tree which maximizes the informative score. This algorithm is called S2+.

Recent structural learning algorithms with *unbounded* treewidth (Scanagatta et al., 2015) can cope with thousands of variables. Yet the unbounded treewidth provides no guarantee about the tractability of the inferences of the learned models. We aim at filling this gap, learning treewidth-bounded Bayesian network models in domains with thousands of variables.

We propose two novel methods for learning Bayesian networks with bounded treewidth. They exploit the fact that any k-tree can be constructed by an iterative procedure that adds one variable at a time. We propose an iterative procedure that, given an order on the variables, builds a DAG $G$ adding one variable at a time. The moral graph of $G$ is ensured to be subgraph of a k-tree. The k-tree is designed as to maximize the score of the resulting DAG. This is a major difference with respect to previous works (Nie et al., 2015, 2016) in which the k-trees were randomly sampled. We propose both an exact and an approximated variant of our algorithm; the latter is necessary to scale to thousands of variables.

We discuss that the search space of the presented algorithms does not span the whole space of bounded-treewidth DAGs. Yet our algorithms consistently outperform the state-of-the-art competitors for structural learning with bounded treewidth. For the first time we present experimental results for structural learning with bounded treewidth for domains involving up to *ten thousand* variables.

Software and supplementary material are available from `http://blip.idsia.ch`.

## 2 Structural learning

Consider the problem of learning the structure of a Bayesian network from a complete data set of $N$ instances $\mathcal{D} = \{D_1, ..., D_N\}$. The set of $n$ categorical random variables is $\mathcal{X} = \{X_1, ..., X_n\}$. The goal is to find the best DAG $G = (V, E)$, where $V$ is the collection of nodes and $E$ is the collection of arcs. $E$ can be represented by the set of parents $\Pi_1, ..., \Pi_n$ of each variable.

Different scores can be used to assess the fit of a DAG; we adopt the *Bayesian information criterion* (or simply BIC). The BIC score is *decomposable*, being constituted by the sum of the scores of the individual variables:

$$\mathrm{BIC}(G) = \sum_{i=1}^{n} \mathrm{BIC}(X_i, \Pi_i) = \sum_{i=1}^{n} \left(\mathrm{LL}(X_i | \Pi_i) + \mathrm{Pen}(X_i, \Pi_i)\right) =$$

$$= \sum_{i=1}^{n} \left(\sum_{\pi \in |\Pi_i|, x \in |X_i|} N_{x,\pi} \hat{\theta}_{x|\pi} - \frac{\log N}{2}(|X_i| - 1)(|\Pi_i|)\right)$$

where $\hat{\theta}_{x|\pi}$ is the maximum likelihood estimate of the conditional probability $P(X_i = x | \Pi_i = \pi)$, $N_{x,\pi}$ represents the number of times $(X = x \wedge \Pi_i = \pi)$ appears in the data set, and $|\cdot|$ indicates the size of the Cartesian product space of the variables given as argument. Thus $|X_i|$ is the number of states of $X_i$ and $|\Pi_i|$ is the product of the number of states of the parents of $X_i$.

Exploiting decomposability, we first identify independently for each variable a list of candidate parent sets (parent set identification). Later, we select for each node the parent set that yields the highest-scoring treewidth-bounded DAG (structure optimization).

## 2.1 Treewidth and $k$-trees

We illustrate the concept of treewidth following the notation of Elidan and Gould (2009). We denote an undirected graph as $H = (V, E)$ where $V$ is the vertex set and $E$ is the edge set. A *tree decomposition* of $H$ is a pair $(\mathcal{C}, \mathcal{T})$ where $\mathcal{C} = \{C_1, C_2, ..., C_m\}$ is a collection of subsets of $V$ and $T$ is a tree on $\mathcal{C}$, so that:

- $\cup_{i=1}^{m} C_i = V$;
- for every edge which connects the vertices $v_1$ and $v_2$, there is a subset $C_i$ which contains both $v_1$ and $v_2$;
- for all $i, j, k$ in $\{1, 2, ..m\}$ if $C_j$ is on the path between $C_i$ and $C_k$ in $\mathcal{T}$ then $C_i \cap C_k \subseteq C_j$.

The width of a tree decomposition is $\max(|C_i|) - 1$ where $|C_i|$ is the number of vertices in $C_i$. The treewidth of $H$ is the minimum width among all possible tree decompositions of $G$.

The treewidth can be equivalently defined in terms of a triangulation of $H$. A triangulated graph is an undirected graph in which every cycle of length greater than three contains a chord. The treewidth of a triangulated graph is the size of the maximal clique of the graph minus one. The treewidth of $H$ is the minimum treewidth over all the possible triangulations of $H$.

The treewidth of a Bayesian network is characterized with respect to all possible triangulations of its moral graph. The moral graph $M$ of a DAG is an undirected graph that includes an edge $i - j$ for every edge $i \rightarrow j$ in the DAG and an edge $p - q$ for every pair of edges $p \rightarrow i$, $q \rightarrow i$ in the DAG. The treewidth of a DAG is the minimum treewidth over all the possible triangulations of its moral graph $M$. Thus the maximal clique of any moralized triangulation of $G$ is an upper bound on the treewidth of the model.

$k$**-trees**   An undirected graph $T_k = (V, E)$ is a $k$-tree if it is a *maximal* graph of tree-width $k$: any edge added to $T_k$ increases its treewidth. A $k$-tree is inductively defined as follows (Patil, 1986). Consider a $(k + 1)$-clique, namely a complete graph with $k + 1$ nodes. A $(k + 1)$-clique is a $k$-tree. A $(k + 1)$-clique can be decomposed into multiple $k$-cliques. Let us denote by $z$ a node not yet included in the list of vertices $V$. Then the graph obtained by connecting $z$ to every node of a $k$-clique of $T_k$ is also a $k$-tree. The treewidth of any subgraph of a $k$-tree (*partial $k$-tree*) is bounded by $k$. Thus a DAG whose moral graph is subgraph of a $k$-tree has treewidth bounded by $k$.

## 3   Incremental treewidth-bounded structure learning

Our approach for the structure optimization task proceeds by repeatedly sampling an order $\prec$ over the variables and then identifying the highest-scoring DAG with bounded treewidth consistent with the order. An effective approach for structural learning based on order sampling has been introduced by Teyssier and Koller (2012); however it does not enforce any treewidth constraint.

The size of the search space of orders is $n!$; this is smaller than the search space of the k-trees, $O(e^{n \log(nk)})$. Once the order $\prec$ is sampled, we incrementally learn the DAG. At each iteration the moralization of the DAG is by design a subgraph of a $k$-tree. The treewidth of the DAG eventually obtained is thus bounded by $k$. The algorithm proceeds as follows.

**Initialization**   The initial k-tree $\mathcal{K}_{k+1}$ is constituted by the complete clique over the first $k + 1$ variables in the order. The initial DAG $\mathcal{G}_{k+1}$ is learned over the same $k + 1$ variables. Since $(k + 1)$ is a tractable number of variables, we exactly learn $\mathcal{G}_{k+1}$ adopting the method of Cussens (2011). The moral graph of $\mathcal{G}_{k+1}$ is a subgraph of $\mathcal{K}_{k+1}$ and thus $\mathcal{G}_{k+1}$ has bounded treewidth.

**Addition of the subsequent nodes**   We then iteratively add each remaining variable in the order. Consider the next variable in the order, $X_{\prec i}$, where $i \in \{k + 2, ..., n\}$. Let us denote by $\mathcal{G}_{i-1}$ and $\mathcal{K}_{i-1}$ the DAG and the k-tree which have to be updated by adding $X_{\prec i}$. We add $X_{\prec i}$ to $\mathcal{G}_{i-1}$, constraining its parent set $\Pi_{\prec i}$ to be a $k$-clique (or a subset of) in $\mathcal{K}_{i-1}$. This yields the updated DAG $\mathcal{G}_i$. We then update the k-tree, connecting $X_{\prec i}$ to such $k$-clique. This yields the k-tree $\mathcal{K}_i$; it contains an additional $k + 1$-clique compared to $\mathcal{K}_{i-1}$. By construction, $\mathcal{K}_i$ is also a $k$-tree. The moral graph of $\mathcal{G}_i$ cannot add arc outside this $(k + 1)$-clique; thus it is a subgraph of $\mathcal{K}_i$.

**Pruning orders**   The initial k-tree $\mathcal{K}_{k+1}$ and the initial DAG $\mathcal{G}_{k+1}$ depend on which are the first $k + 1$ variables in the order, but not on their *relative* positions. Thus all the orders which differ only

as for the relative position of the first $k + 1$ elements are *equivalent* for our algorithm: they yield the same $\mathcal{K}_{k+1}$ and $\mathcal{G}_{k+1}$. Thus once we sample an order and perform structural learning, we prune the $(k + 1)! - 1$ orders which are equivalent to the current one.

In order to choose the parent set to be assigned to each variable added to the graph we propose two algorithms: k-A* and k-G.

## 3.1  k-A*

We formulate the problem as a *shortest path finding* problem. We define each state as a step towards the completion of the structure, where a new variable is added to the DAG $\mathcal{G}$. Given $X_{\prec i}$ the variable assigned in the state $S$, we define a successor state of $S$ for each $k$-clique to which we can link $X_{\prec i+1}$. The approach to solve the problem is based on a path-finding A* search, with cost function for state $S$ defined as $f(S) = g(S) + h(S)$. The goal is to find the state which minimizes $f(S)$ once all variables have been assigned.

We define $g(S)$ and $h(S)$ as:

$$g(S) = \sum_{j=0}^{i} score(X_{\prec j}, \Pi_{\prec j}), \qquad\qquad h(S) = \sum_{j=i+1}^{n} best(X_{\prec j}).$$

$g(S)$ is the cost from the initial state to $S$; it corresponds to the sum of scores of the already assigned parent sets.

$h(S)$ is the estimated cost from $S$ to the goal. It is the sum of the best assignable parent sets for the remaining variables. Variable $X_a$ can have $X_b$ as parent only if $X_b \prec X_a$.

The A* approach requires the $h$ function to be *admissible*. The function $h$ is admissible if the estimated cost is never greater than the true cost to the goal state. Our approach satisfies this property since the true cost of each step (score of chosen parent set for $X_{\prec i+1}$) is always equal to or greater than the estimated one (the score of the best selectable parent set for $X_{\prec i+1}$).

The previous discussion implies that $h$ is *consistent*, meaning that for any state $S$ and its successor $T$, $h(S) \leq h(T) + c(S, T)$, where $c(S, T)$ is the cost of the edges added in $T$. The function $f$ is monotonically non-decreasing on any path, and the algorithm is guaranteed to find the optimal path as long as the goal state is reachable. Additionally there is no need to process a node more than once, as no node will be explored a second time with a lower cost.

## 3.2  k-G

A very high number of variables might prevent the use of k-A*. For those cases we propose k-G as a greedy alternative approach, which chooses at each step the best local parent set. Given the set of existing $k$-clique in $\mathcal{K}$ as $\mathcal{K}_C$, we choose as parent set for $X_{\prec i}$:

$$\Pi_{X_{\prec i}} = \underset{\pi \subset c, c \in \mathcal{K}_C}{\operatorname{argmax}} \; score(\pi).$$

## 3.3  Space of learnable DAGs

A *reverse topological order* is an order $\{v_1, ... v_n\}$ over the vertexes $V$ of a DAG in which each $v_i$ appears before its parents $\Pi_i$. The search space of our algorithms is restricted to the DAGs whose reverse topological order, when used as variable elimination order, has treewidth $k$. This prevents recovering DAGs which have bounded treewidth but lack this property.

We start by proving by induction that the reverse topological order has treewidth $k$ in the DAGs recovered by our algorithms. Consider the incremental construction of the DAG previously discussed. The initial DAG $\mathcal{G}_{k+1}$ is induced over $k + 1$ variables; thus every elimination ordering has treewidth bounded by $k$.

For the inductive case, assume that $\mathcal{G}_{i-1}$ satisfies the property. Consider the next variable in the order, $X_{\prec_i}$, where $i \in \{k + 2, ..., n\}$. Its parent set $\Pi_{\prec_i}$ is a subset of a $k$-clique in $\mathcal{K}_{i-1}$. The only neighbors of $X_{\prec_i}$ in the updated DAG $\mathcal{G}_i$ are its parents $\Pi_{\prec_i}$. Consider performing variable elimination on the moral graph of $\mathcal{G}_i$, using a reverse topological order. Then $X_{\prec_i}$ will be eliminated before $\Pi_{\prec_i}$, without introducing fill-in edges. Thus the treewidth associated to any reverse topological order is bounded by $k$. This property inductively applies to the addition also of the following nodes up to $X_{\prec_n}$.

**Inverted trees** An example of DAG non recoverable by our algorithms is the specific class of polytrees that we call *inverted trees*, that is, DAGs with out-degree equal to one. An inverted tree with $m$ levels and treewidth $k$ can be built as follows. Take the root node (level one) and connect it to $k$ child nodes (level two). Connect each node of level two to $k$ child nodes (level three). Proceed in this way up to the m-th level and then invert the direction of all the arcs.

Figure 1 shows an inverted tree with $k$=2 and $m$=3. It has treewidth two, since its moral graph is constituted by the cliques {A,B,E}, {C,D,F}, {E,F,G}. The treewidth associated to the reverse topological order is instead three, using the order G, F, D, C, E, A, B.

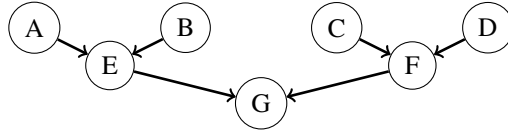

Figure 1: Example of inverted tree.

If we run our algorithms with bounded treewidth $k$=2, it will be unable to recover the actual inverted tree. It will instead identify a high-scoring DAG whose reverse topological order has treewidth 2.

## 4 Experiments

We compare k-A*, k-G, S2, S2+ and TWILP in various experiments. We compare them through an indicator which we call W-score: the percentage of worsening of the BIC score of the selected treewidth-bounded method compared to the score of the Gobnilp solver Cussens (2011). Gobnilp achieves higher scores than the treewidth-bounded methods since it has no limits on the treewidth. Let us denote by $G$ the BIC score achieved by Gobnilp and by $T$ the BIC score obtained by the given treewidth-bounded method. Notice that both $G$ and $T$ are negative. The W-score is $W = \frac{G-T}{G}$. W stands for worsening and thus lower values of $W$ are better. The lowest value of $W$ is zero, while there is no upper bound.

We adopt BIC as a scoring function. The reason is that an algorithm for approximate exploration of the parent sets (Scanagatta et al., 2015) allowing high in-degree even on large domains exists at the moment only for BIC.

### 4.1 Parent set score exploration

Before performing structural learning it is necessary to compute the scores of the candidate parent sets for each node (parent set exploration). The different structural learning methods are then provided with the same score of the parent sets.

A treewidth $k$ implies that one should explore all the parent sets up to size $k$; thus the complexity of parent set exploration increases exponentially with the treewidth. To let the parent set exploration scale efficiently with large treewidths and large number of variables we apply the approach of Scanagatta et al. (2015). It guides the exploration towards the most promising parent sets (with size up to $k$) without scoring them all. This is done on the basis of an approximated score function that is computed in constant time. The actual score of the most promising parent sets is eventually computed. We allow 60 seconds of time for the computation of the scores of the parent set of each variable, in each data set.

### 4.2 Our implementation of S2 and S2+

Here we provide some details of our implementation of S2 and S2+. The second phase of both S2 and S2+ looks for a DAG whose moralization is a subgraph of a chosen k-tree. For this task Nie et al. (2014) adopt an approximate approach based on partial order sampling (Algorithm 2). We found that using Gobnilp for this task yields consistently slightly higher scores; thus we adopt this approach in our implementation. We believe that it is due to the fact that constraining the structure optimization to a subjacent graph of a k-tree results in a small number of allowed arcs for the DAG. Thus our implementation of S2 and S2+ finds the highest-scoring DAG whose moral graph is a subgraph of the provided k-tree.

### 4.3 Learning inverted trees

As already discussed our approach cannot learn an inverted tree with $k$ parents per node if the given bounded treewidth is $k$. In this section we study this worst-case scenario.

We start with treewidth $k = 2$. We consider the number of variables $n \in \{21, 41, 61, 81, 101\}$. For each value of $n$ we generate 5 different inverted trees. To generate as inverted tree we first select a root variable $X$ and add $k$ parents to it as $\Pi_X$; then we continue by randomly choosing a leaf of the graph (at a generic iteration, there are leaves at different distance from $X$) and adding $k$ parents to it, until the the graph contains $n$ variables.

All variables are binary and we sample their conditional probability tables from a Beta(1,1). We sample 10,000 instances from each generated inverted tree.

We then perform structural learning with k-A*, k-G, S2, S2+ and TWILP setting $k = 2$ as limit on the treewidth. We allow each method to run for ten minutes. S2, S2+ and TWILP could in principle recover the true structure, which is prevented to our algorithms. The results are shown in Fig.2. Qualitatively similar results are obtained repeating the experiments with $k = 4$.

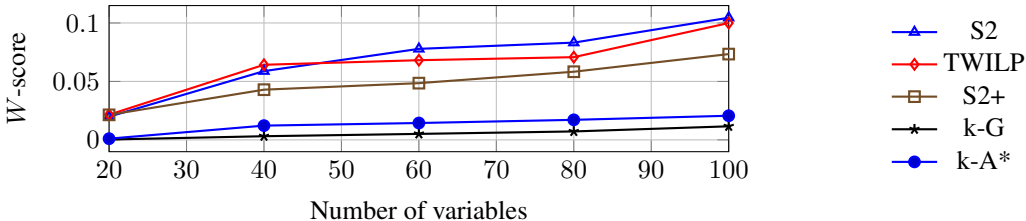

Figure 2: Structural learning results when the actual DAGs are inverted trees ($k$=2). Each point represent the mean W-score over 5 experiments. Lower values of the $W$-score are better.

Despite the unfavorable setting, both k-G and k-A* yield DAGs with higher score than S2, S2+ and TWILP consistently for each value of $n$. For $n = 20$ they found a close approximation to the optimal graph. S2, S2+ and TWILP found different structures, with close score.

Thus the limitation of the space of learnable DAGs does not hurt the performance of k-G and k-A*. In fact S2 could *theoretically* recover the actual DAG, but this is not feasible in practice as it requires a prohibitive number of samples from the space of k-trees. The exact solver TWILP was unable to find the exact solution within the time limit; thus it returned a best solution achieved in the time limit.

|  | S2 | S2+ | k-G | k-A* |
|---|---|---|---|---|
| Iterations | 803150 | 3 | 7176 | 66 |
| Median | -273600 | -267921 | -261648 | -263250 |
| Max | -271484 | -266593 | -258601 | -261474 |

Table 1: Statistics of the solutions yielded by different methods on an inverted tree ($n = 100$, $k = 4$).

We further investigate the differences between methods in Table 1. *Iterations* is the number of proposed solutions; for S2 and S2+ it corresponds to the number of explored k-trees, while for k-G and k-A* it corresponds to the number of explored orders. During the execution, S2 samples almost one million k-trees. Yet it yields the lowest-scoring DAGs among the different methods. This can be explained considering that a randomly sampled k-tree has a low chance to cover a high-scoring DAG. S2+ recovers only a few k-trees, but their scores are higher than those of S2. Thus the informative score is effective at driving the search for good k-trees; yet it does not scale on large data sets as we will see later. As for our methods, k-G samples a larger number of orders than k-A* does and this allows it to achieve higher scores, even if it sub-optimally deals with each single order. Such statistics show a similar pattern also in the next experiments.

| DATASET | VAR. | GOBNILP | TWILP | S2 | S2+ | k-G | k-A* |
|---|---|---|---|---|---|---|---|
| nursery | 9 | -72159 | −**72159** | −**72159** | −**72159** | −**72159** | −**72159** |
| breast | 10 | -2698 | −**2698** | −**2698** | −**2698** | −**2698** | −**2698** |
| housing | 14 | -3185 | -3213 | -3252 | -3247 | -3206 | −**3203** |
| adult | 15 | -200142 | -200340 | -201235 | -200926 | -200431 | −**200363** |
| letter | 17 | -181748 | -190086 | -189539 | -186815 | -183369 | −**183241** |
| zoo | 17 | -608 | -620 | -620 | -619 | -615 | −**613** |
| mushroom | 22 | -53104 | -68298 | -68670 | -64769 | -57021 | −**55785** |
| wdbc | 31 | -6919 | -7190 | -7213 | -7209 | -7109 | −**7088** |
| audio | 62 | -2173 | -2277 | -2283 | -2208 | -2201 | −**2185** |
| community | 100 | -77555 | | -107252 | -88350 | -82633 | −**82003** |
| hill | 100 | -1277 | | -1641 | -1427 | -1284 | −**1279** |

Table 2: Comparison of the BIC scores yielded by different algorithms on the data sets analyzed by Nie et al. (2016). The highest-scoring solution with limited treewidth is boldfaced. In the first column we report the score obtained by Gobnilp without bound on the treewidth.

## 4.4 Small data sets

We now present experiments on the data sets considered by Nie et al. (2016). They involve up to 100 variables. We set the bounded treewidth to $k = 4$. We allow each method to run for ten minutes. We perform 10 experiments on each data set and we report the median scores in Table 2.

On the smallest data sets all methods (including Gobnilp) achieve the same score. As the data sets becomes larger, both k-A* and k-G achieve higher scores than S2, S2+ and TWILP (which does not achieve the exact solution). Between our two novel algorithms, k-A* has a slight advantage over k-G.

## 4.5 Large data sets

We now consider 10 large data sets ($100 \leq n \leq 400$) listed in Table 3. We no longer run TWILP, as it is unable to handle this number of variables.

| Data set | $n$ | Data set | $n$ | Data set | $n$ | Data set | $n$ | Data set | $n$ |
|---|---|---|---|---|---|---|---|---|---|
| Audio | 100 | Netflix | 100 | Retail | 135 | Andes | 223 | Pumsb-star | 163 |
| Jester | 100 | Accidents | 111 | Kosarek | 190 | MSWeb | 294 | DNA | 180 |

Table 3: *Large* data sets sorted according to the number of variables.

| | k-A* | S2 | S2+ |
|---|---|---|---|
| k-G | **29/20/24** | **30/30/29** | **30/30/30** |
| k-A* | | **29/27/20** | **29/27/21** |
| S2 | | | 12/13/**30** |

Table 4: Result on the 30 experiments on large data sets. Each cell report how many times the row algorithm yields a higher score than the column algorithm for treewidth 2/5/8. For instance k-G wins on all the 30 data sets against S2+ for each considered treewidth.

We consider the following treewidths: $k \in \{2, 5, 8\}$. We split each data set randomly into three subsets. Thus for each treewidth we run 10·3=30 structural learning experiments.

We let each method run for one hour. For S2+, we adopt a more favorable approach, allowing it to run for one hour; if after one hour the first k-tree was not yet solved, we allow it to run until it has solved the first k-tree.

In Table 4 we report how many times each method wins against another for each treewidth, out of 30 experiments. The entries are boldfaced when the number of victories of an algorithm over another is statistically significant (p-value <0.05) according to the sign-test. Consistently for any chosen treewidth, k-G is significantly better than any competitor, including k-A*; moreover, k-A* is significantly better than both S2 and S2+.

This can be explained by considering that k-G explores more orders than k-A*, as for a given order it only finds an approximate solution. The results suggest that it is more important to explore many orders instead of obtaining the optimal DAG given an order.

## 4.6 Very large data sets

Eventually we consider 14 very large data sets, containing between 400 and 10000 variables. We split each algorithm in three subsets. We thus perform 14·3=42 structural learning experiments with each algorithm.

We include three randomly-generated synthetic data sets containing 2000, 4000 and 10000 variables respectively. These networks have been generated using the software BNGenerator [4]. Each variable has a number of states randomly drawn from 2 to 4 and a number of parents randomly drawn from 0 to 6.

| Data set | $n$ | Data set | $n$ | Data set | $n$ | Data set | $n$ | Data set | $n$ |
|---|---|---|---|---|---|---|---|---|---|
| Diabets | 413 | EachMovie | 500 | Reuters-52 | 889 | BBC | 1058 | R4 | 4000 |
| Pigs | 441 | Link | 724 | C20NG | 910 | Ad | 1556 | R10 | 10000 |
| Book | 500 | WebKB | 839 | Munin | 1041 | R2 | 2000 | | |

Table 5: Very large data sets sorted according to the number $n$ of variables.

We let each method run for one hour. The only two algorithms able to cope with these data sets are k-G and S2. For all the experiments, both k-A* and S2+ fail to find even a single solution in the allowed time limit; we verified this is not due to memory issues. Among them, k-G wins 42 times out of 42; this dominance is clearly significant. This result is consistently found under each choice of treewidth ($k = 2, 5, 8$). On average, the improvement of k-G over S2 fills about 60% of the gap which separates S2 from the unbounded solver.

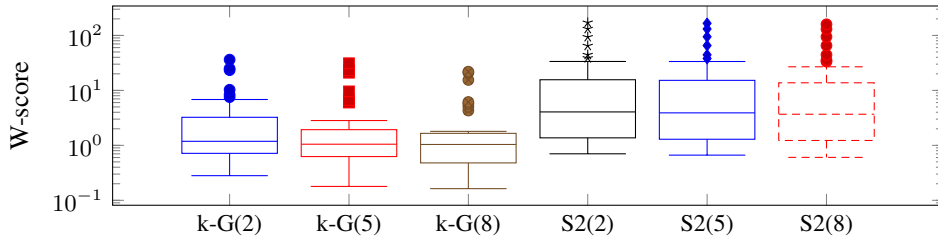

Figure 3: Boxplots of the W-scores, summarizing the results over 14·3=42 structural learning experiments on very large data sets. Lower W-scores are better. The y-axis is shown in logarithmic scale. In the label of the x-axis we also report the adopted treewidth for each method: 2, 5 or 8.

The W-scores of such 42 structural learning experiments are summarized in Figure 3. For both S2 and k-G, a larger treewidth allows to recover a higher-scoring graph. In turn this decreases the W-score. However k-G scales better than S2 with respect to the treewidth; its W-score decreases more sharply with the treewidth. For S2, the difference between the treewidth seems negligible from the figure. This is due to the fact that the graph learned are actually sparse.

Further experimental documentation is available, including how the score achieved by the algorithms evolve with time, are available from `http://blip.idsia.ch`.

## 5 Conclusions

Our novel approaches for treewidth-bounded structure learning scale effectively with both in the number of variables and the treewidth, outperforming the competitors.

**Acknowledgments**

Work partially supported by the Swiss NSF grants 200021_146606 / 1 and IZKSZ2_162188.

## Footnotes

*Istituto Dalle Molle di studi sull'Intelligenza Artificiale (IDSIA)

†Scuola universitaria professionale della Svizzera italiana (SUPSI)

‡Università della Svizzera italiana (USI)

[4]`http://sites.poli.usp.br/pmr/ltd/Software/BNGenerator/`

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
