[Reviews · NeurIPS 2016]

Reviewer 1

Summary

This paper uses k-trees to learn bounded tree-width Bayesian networks. The proposed method is built on Nie et al. 2016, which incrementally grows the k-tree and the BN as the subgraph of the k-tree. The parent set of a variable in the BN is restricted by the clique of k nodes that the variable is connected to in the k-tree. To find this clique of k nodes, the authors use A* algorithm similar to Nie et al. 2016 but with a different score function that is consistent with the overall objective function. They also propose a greedy search approach instead of A* to increase the scalability.

Qualitative Assessment

The proposed method is very similar to previous work by Nie et al. -- both use k-trees to search for low-treewidth Bayesian networks, both start with a randomly chosen initial clique, and both propose using an A* method for finding the best tree. The differences are that Nie et al. score k-trees using a mutual information score and use BDeu for choosing the final consistent Bayesian network, while this paper proposes using BIC and incrementally building the Bayesian network along with the k-tree, using the BN to score the k-tree. This paper also includes the additional restriction that the complete variable (partial) order is chosen randomly, while in Nie et al. only the initial clique is chosen randomly. The main justification for these differences is the ability to scale to large treewidths. However, in the experiments, the previous S2 algorithm also can scale to large treewidths. The experimental results look promising, showing that the new method can optimize BIC better (given a treewidth constraint and a particular bound on running time). Comments: - Why use BIC instead of BDeu, as in previous work? As the authors note, this makes their results incomparable to previous results; what are the advantages that outweigh this obvious disadvantage? - The score function in g(S) (line 135) is not explicitly defined. Do you mean BIC? - It would be helpful to see the performance of all methods with different time limits (e.g., 1 minute, 1 hour, 10 hours). - Many bounded-treewidth methods exist in the literature, some for learning chain graphs and some for learning BNs; the results would be more convincing with additional baselines. (I would be interested in seeing test set log-likelihood as well, though I understand that the stated goal of these algorithms is to optimize a particular criterion as well as possible, not to evaluate generalization.) - The paper is clear for the most part, but the presentation could be improved. For example, it would help to show the full table of results for the large datasets, not just a boxplot and counts of which algorithm was better. Supplementary material could be used for this if it will not fit in the main paper. Minor: - The sentences at line 45 and 46 are redundant. Maybe authors want to merge them together. - The space of learnable DAGs is not well explored. It would be interesting to see some experiments that show the edge recovery potential of the proposed algorithm by generating samples from a different set of synthetic graphs.

Confidence in this Review

3-Expert (read the paper in detail, know the area, quite certain of my opinion)


Reviewer 2

Summary

The paper presents a method for learning BNs of bounded treewidth which works even with very many variables. The algorithm repeatedly samples an order over the variables and then incrementally constructs the BN. Thorough empirical comparisons with a number of other approaches are made.

Qualitative Assessment

This is a high quality paper. The methods are clearly described and the empirical comparisons thorough and they provide support for the effectiveness of the authors' approach. There is a useful section on what BNs the presented approach *cannot* learn. In the empirical section this limitation is explored (section 4.3) and it is found that even when the true BN is unlearnable k-G and k-A* do well. I was left wondering what motivated the design of k-G/k-A*. OK, they work well, but why? Some theoretical work on how far the output of k-G/k-A* typically is from an optimal (bounded-treewidth) BN would be welcome. typo: line 184 higher score -> higher scores

Confidence in this Review

3-Expert (read the paper in detail, know the area, quite certain of my opinion)


Reviewer 3

Summary

This paper presents new heuristics for learning bounded treewidth Bayesian networks.

Qualitative Assessment

The problem with the algorithms for learning bounded treewidth Bayesian networks is that they are usually too slow to be practical. This paper introduces a new heuristic that scales up better than the previous methods. While the proposed method has no theoretical guarantees and it only searches within a small subset of bounded treewidth graphs, it improves state-of-the-art and thus it may have impact. The paper is easy to read and as far as I can see correct. The presented algorithms have a weakness that their search space is not the whole space of bounded treewidth DAGs. This limitation should be mentioned already in the introduction. Now it is hidden in Section 3.3 and from line 103 one gets an impression that such a limitation doesn't exist. In Figure 2, k-G and k-A* seem to have zero W-score for n=21 which implies that they found the optimal graph. But that shouldn't be possible given their limitation? Furthermore, S2, S2+ and TWILP seem to all produce the same suboptimal result. This seems like a strange coincidence. Line 92: G -> M Line 155: vertexes -> vertices Teyssier & Koller paper is from UAI 2005

Confidence in this Review

3-Expert (read the paper in detail, know the area, quite certain of my opinion)


Reviewer 4

Summary

The paper introduced a novel method of Bayesian network structure learning. The new method is built upon the famous Order Search method, by taking consideration of the tree-width during the learning. Numerical experiments show that the new approach wins other state-of-the-art methods on learning bounded tree-width Bayesian network,

Qualitative Assessment

The paper is well organized and the mathematics notations are easy to understand. The new method, as reported in the experiments, wins other methods in the task of learning bounded tree-width Bayesian network. The experiments are comprehensive and supportive. A few minor suggestions would be: - Is there any specific reason that the author adopted BIC score instead of BDeu? It seems not hard to plug in the BDeu instead of BIC score in the parent scoring phase and do the same experiments. - The algorithm randomly samples order and do DAG learning based on each sampled orders. I think it would be desirable to report the number of orders sampled in the experiments (say in the computation time of 10 minutes). That would give the reader more intuition I think. - In line 246, the author states that the results of Nie 2016’s results are not comparable because different scores. I don’t think that is a problem because both learning outputs would be a structure and we simply need to compute the score of two structures and compare. Am I correct? - What is the computing budget for very large dataset? Also 1 hour as the large datasets case? I think a plot of the learning curve (graph score vs computing time) would be helpful to clear some doubts that S2 and S2+ might need some more time to learn a better structure. For example, I think for the small dataset maybe we can accept for half an hour to learn and for large one maybe several hours?

Confidence in this Review

2-Confident (read it all; understood it all reasonably well)


Reviewer 5

Summary

In this paper, the authors propose an approach to learn a treewidth-bounded Bayesian Network which represent a given dataset. The returned treewidth-bounded BN structure approximately maximizes the BIC criterion (given the data). The BN structure learning problem is rather classical, has received a lot of attention. The problem of finding a treewidth-bounded BN is very important since bounded treewidth implies more efficient inference. This problem has also attracted a lot of attention recently. After describing the current state of the art on bounded-treewidth BN inference (I think the description of the state of the art is complete), the authors focus on two approaches proposed by Nie et al. 2015, 2016. In these approaches, the set of k-trees are explored, either randmoly, or using a A* algorithm, to optimize "informative score". The current approach is similar to Nie et al., 2016, except that a BIC score is "maximized". Two approximate algorithms are proposed, k-A* and k-G (a greedy maximization algorithm) and experimentally compared with the approaches of Nie et al., 2015-2016. It is found that k-A* consistently outperforms the existing approaches for small problems, while k_G does so for larger problems.

Qualitative Assessment

One could object that the originality with respect to the state-of-the-art approaches is limited since the difference with the Nie et al. approaches are that : (i) a different score is maximised and (ii) a more informed exploration of the set of k-trees is used. However, these differences are enough to improve over the Nie et al. approaches. The methods described could have a practical impact in the field of "learning", given that tree-width-bounded graphical models are useful general-purpose representations of data. Experiments are complete, appropriate and convincing. Overall, the paper is well-written, solid and provides sufficient experiments to assess the validity of the contribution. Given the clarity and solidity of the paper and despite my slight reservation on originality, my opinion is that the contribution is enough for the paper to be published.

Confidence in this Review

2-Confident (read it all; understood it all reasonably well)


Reviewer 6

Summary

The paper presents a new approach for learning the structure of Bayesian networks with bounded treewidth from data. Treewidth bounded graphical models such as k-trees are important because inference is tractable which is very practical when learning the parameters of the model from data. Two structure learning algorithms are introduced for learning k-trees: an exact one (k-A*) which is based on best-first search, and a greedy approximation (k-G) which is able to scale to datasets containing up to 10000 variables. The empirical evaluation of several benchmarks ranging from easy to very difficult instances shows that the new algorithms are competitive and in many cases they outperform considerably (in terms of solution quality) existing state of the art approaches for learning bounded treewidth Bayesian networks.

Qualitative Assessment

I found the paper well written and organized, and therefore relatively easy to follow. Most of the concepts are introduced and discussed in detail. The quality of presentation is good in general. Maybe a simple running example that shows how k-A* works would help better understand its details. Overall, I think the paper provides a significant contribution to the graphical models community, adding to the collection of structure learning algorithms. Although the contribution seems to focus on k-trees only (which is a particular class of bounded treewidth graphical models), to what extent the method/idea is applicable to other classes of bounded treewidth graphical models. Also, there are practical situations where learning/using an undirected model is desirable, so what would be the requirements to extend the proposed approach to handle undirected graphical models.

Confidence in this Review

3-Expert (read the paper in detail, know the area, quite certain of my opinion)